EQUITY, DIVERSITY AND INCLUSION

# A guide for writing anti-racist tenure and promotion letters

**Abstract** In a two-page tenure letter, senior faculty can make or break a career. This power has an outsized impact on Black academics and other scholars with marginalized identities, who are awarded tenure at lower rates than their white colleagues. We suggest that this difference in tenure rates is due to an implicit, overly narrow definition of academic excellence that does not recognize all the contributions that Black scholars make to their departments, institutions and academia in general. These unrecognized contributions include the (often invisible) burdens of mentoring and representation that these scholars bear disproportionately. Here we propose a set of practical steps for writing inclusive, anti-racist tenure letters, including what to do before writing the letter, what to include (and not include) in the letter itself, and what to do after writing the letter to further support the candidate seeking tenure. We are a group of mostly non-Black academics in science, technology, engineering and mathematics (STEM) based in the United States who are learning about and working toward Black liberation in academia; we hope these recommendations will help ongoing efforts to move toward an inclusive academia that appreciates and rewards diverse ways of doing, learning and knowing.

**THE A4BL ANTI-RACIST TENURE LETTER WORKING GROUP\***

**\*For correspondence:**
pearisbellamy@ufl.edu;chaynes@
umn.edu;lmartin2@kumc.edu;
spmirabile@smcm.edu;yael@
princeton.edu;rosej8@wwu.edu;
rachel.ross@einsteinmed.edu

**Group author details:**
The A4BL Anti-racist Tenure
Letter Working Group See page 8

**Competing interest:** The authors declare that no competing interests exist.

## Introduction

Senior faculty can make, shape, and – unfortunately – break the careers of junior faculty, especially within the tenure review process. Black scholars, already severely underrepresented as junior faculty, are promoted with tenure at a lower rate than their white colleagues (*Frazier, 2011*; *Herbert, 2012*; *Stewart, 2012*; *US Department of Education, 2017*). And not because they are not as 'good'. If anything, Black scholars have had to overcome more hurdles than their white colleagues to become assistant professors (*Diep, 2020*; *Frazier, 2011*), surviving and excelling in long and unforgiving training and selection processes that are biased against them. Why, then, are they not being promoted?

We believe this is primarily due to two factors that tenure letter writers can intentionally work to overturn: (i) a narrow definition of 'excellence' that encompasses only some forms of contribution to academia, and (ii) the unrecognized invisible burdens of mentoring and representation that fall disproportionately on the shoulders of Black faculty (and more so on Black women, who hold doubly marginalized, intersectional identities).

Here we put forth a set of recommendations for writing inclusive, anti-racist, tenure letters that allow letter-writers to counteract these two factors by adopting an expansive view of how scholars contribute to academia, and by promoting the academic culture we all want to support. Throughout, we will refer specifically to Black faculty, as we are writing from a US-centric vantage point on academia, where anti-Black racism imposes myriad, interacting harms on Black scholars (*Bell et al., 2021*). However, the issues we discuss and recommendations we make apply to indigenous scholars, scholars of Color, LGBTQ+ scholars, scholars with disabilities, first-generation scholars, scholars from low-income backgrounds, and, in most academic fields, also women. We emphasize that our recommendations by no means 'lower the bar' of tenure standards in academia. To the contrary, recognizing the full ways that historically excluded scholars excel and help create an excellent academic culture can help raise the standards across academia.

### Academia and 'white supremacy culture'

The academic profession was developed to have strict boundaries about who is allowed to be a

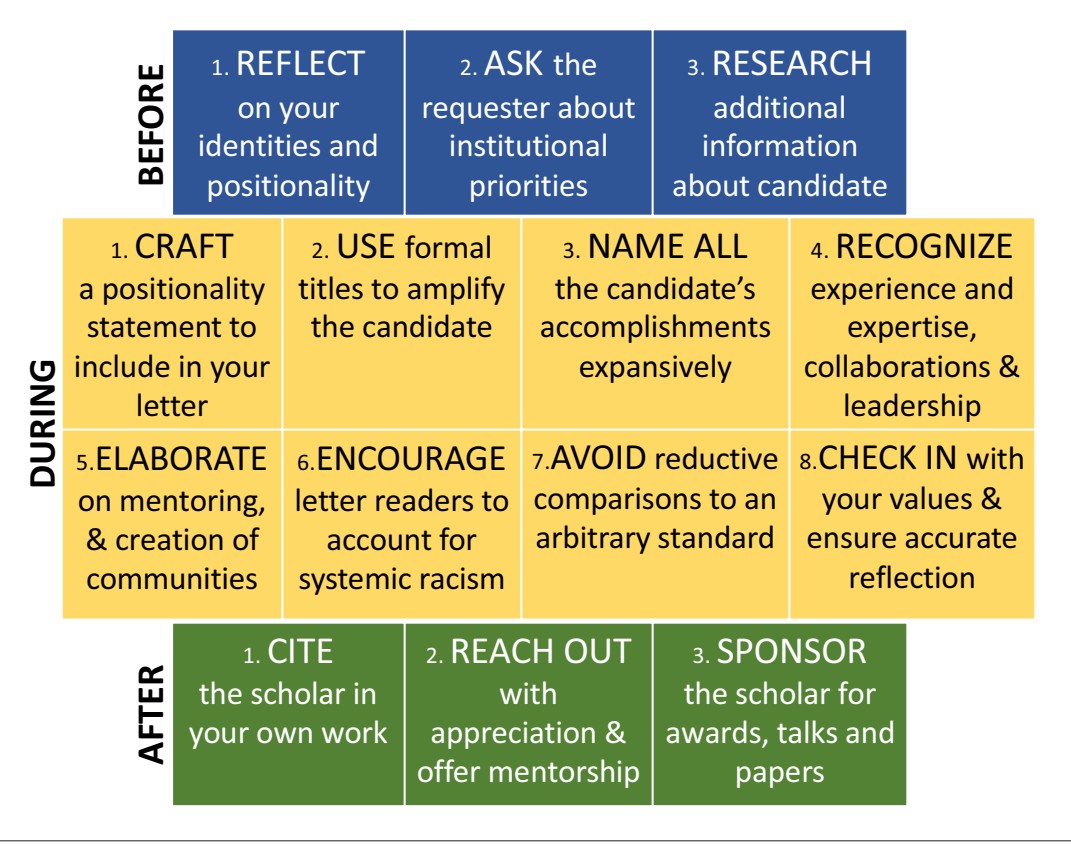

**Figure 1.** Recommendations for writing anti-racist tenure and promotion letters. A set of guidelines for what to do before writing the letter (blue boxes), what to include and not include in the letter itself (yellow boxes), and what to do after writing the letter to further support the candidate (green boxes).

part of the profession and what work is valued. Certain values and qualities (such as objectivity, linearity, generalizability, and detachment) have long been the standard for evaluating academic rigor and granting membership into this highly selective profession (*Gonzales, 2018*). Thus, Black scholars engaging in community-focused research and racial justice work are often evaluated negatively because their work is viewed as not being objective and their advocacy is viewed as problematic (*Gonzales, 2018*). Indeed, Black scholars are overrepresented in topics related to racial discrimination and African studies, but often cited less than scholars from other groups (*Kozlowski et al., 2022*). Therefore, in evaluation processes, like tenure and promotion, their research is often undervalued by evaluators embedded in what Dr. Tema Okun describes as 'white supremacy culture' (*Okun, 1999*; *Okun, 2021*).

According to Okun, "White supremacy culture is the widespread ideology baked into the beliefs, values, norms, and standards of our groups (many if not most of them), our communities, our towns, our states, our nation, teaching us both overtly and covertly that whiteness holds value, whiteness is value" (https://www.whitesupremacyculture.info/what-is-it.html). Whether we are aware of it or not, academic culture is steeped in the beliefs and values Okun associates with 'white supremacy culture', including:

- Perfectionism: the belief that there is 'one right way' to do things and a false sense that we can be objective, and that mistakes are personal.
- Quantity over quality: valuing things that can be measured – publications, grant money – more highly than processes that are harder to quantify (e.g., mentoring relationships, morale).
- Individualism: de-emphasis of team-work and collaboration and over-emphasis on individual achievement and competition.
- Defensiveness: a tendency to protect current systems of power at the expense of hearing new ideas; perceiving criticisms as threats.
- Sense of urgency: an imposed sense of urgency makes it difficult to take time to

be inclusive and to reflect on and learn from mistakes, and draws attention away from truly urgent work for racial justice regardless of academic field.

These values are not a necessity in academia. Most of us, having been trained in this culture for years, may not even recognize these invisible but ever-present 'rules of the game' despite the fact that these rules limit creativity and inclusion. Naming these values as ones we have adopted makes clear that they are not axioms of academic culture. There are alternatives. Those of us committed to disrupting this implicit and harmful culture have a right and obligation to actively promote an academic community that recognizes and benefits from the expertise of all people who participate in academia. One way to accomplish this on an individual level is to reconsider how we write letters of assessment, including tenure and promotion letters, so that they embody the cultural shift we'd like to see.

### What are anti-racist tenure and promotion letters?

Anti-racist tenure and promotion letters provide an avenue of intervention and advocacy to challenge the exclusionary and harmful aspects of academia. In evaluating scholarship that does not necessarily conform to 'white supremacy culture' values, we must recognize that our personal biases influence both our scientific practice and our tendency to uphold these values of scientific pursuit. If we are to move beyond these exclusionary practices, we must recognize these biases in all of our academic practices and value other knowledge systems beyond that of the 'traditional' epistemology of science.

Academia's embodiment of 'white supremacy culture' diminishes the learning environment in our institutions, limits vital representation of Black scholars and their scholarship, and harms Black faculty wellness and opportunities for advancement (*Davis, 2021*; *Diep, 2020*; *Mosley et al., 2021*; *Bell et al., 2021*). These harms pervade all aspects of academia and are particularly well-documented for the tenure and promotion process (*Frazier, 2011*; *Jones et al., 2015*), where examples include lack of recognition for increased service load (*Gewin, 2020*; *Guarino and Borden, 2017*; *Hirshfield and Joseph, 2012*; *Jimenez et al., 2019*; *McCluney and Rabelo, 2019*; *Moore, 2017*, *Social Sciences Feminist Network Research Interest Group, 2017*), marginalization of research (*Padilla, 1994*;

*Settles et al., 2022*), and lower rates of publication and funding (*Taffe and Gilpin, 2021*).

While critical reforms have been identified (e.g., Roadmap for Racial Equity at UNC Chapel Hill) and some institutions – such as, Indiana University–Purdue University Indianapolis and University of Oregon – are pursuing structural remedies to some of these problems (*Truong, 2021*), many scholars who wish to support anti-racist reforms of academia are unable to individually effect such changes in their institutions. Indeed, even scholars who have successfully navigated the tenure process and hold relatively powerful positions in their institutions and disciplines likely still lack the power to change the policies and procedures that harm Black scholars. However, such successful scholars are regularly asked to serve as external evaluators in the tenure process at peer institutions; and it is in this process that individual scholars committed to anti-racism and to an expansive and inclusive academic culture can have a substantial impact on the career trajectories of Black scholars.

Writing tenure/promotion letters is an excellent opportunity to push back against traditional, narrow criteria for promotion and toward a more holistic view of scholarly contributions (for example, see *Renwick et al., 2020*). Although previous guidelines for writing tenure and promotion letters have been proposed (*Goldman, 2016*; *Goldman, 2017*; *Kong et al., 2021*; *Strassman, 2016*; *Female Science Professor, 2014*), they often embrace and perpetuate an academic culture focused on a traditional, narrow understanding of scholarship. Writing a tenure letter purely from one's own lived experience and expectations can also inadvertently introduce bias and undermine the success of the candidate, as most letter writers have not experienced academia as a racially marginalized scholar.

The current system is a vicious positive feedback cycle: letters written by the majority of faculty (i.e., based on the tacitly accepted values of white men) reify the current culture and squander the opportunity to intentionally recognize its limitations and expand beyond it. The goal of the guidelines below is to bring into the tenure and promotion process a holistic, inclusive, and equitable understanding of what successful scholarship can look like and who can be a successful scholar. We intend to broaden recommenders' critical awareness of the scope of scholarship and scholarly activities that are often unrecognized and that fall disproportionately on the shoulders of Black scholars. And we

encourage recommenders to frame and describe such work for *all* tenure/promotion candidates, not only Black or other marginalized candidates, so as to align our evaluation criteria with our stated values.

## Authors' positionality statement

To allow readers to best contextualize our recommendations, we would like to explicitly acknowledge our positionality and limitations in putting together the below recommendations. We are a group of mostly non-Black academics in STEM fields, most of us women, who are learning about and working toward Black liberation in academia. We participated in the 2020 and 2021 "Academics for Black Survival and Wellness (A4BL)" courses envisioned and implemented by Dr. Della Mosley and Pearis Bellamy, with shared wisdom from a large number of anti-Black-racism scholars. This set of recommendations arose from a working group formed during the 2021 course.

We recognize that our project represents an incremental challenge to a system in urgent need of major transformations regarding diversity, equity, and inclusion. When contemplating how we could work toward creating a just, equitable, inclusive, and diverse academia, we recognized many limits to our individual and collective knowledge, experiences, and power. This recognition motivated our interest in identifying areas where a single person has an outsized impact on the career trajectory of Black scholars. We therefore offer these recommendations that are based on what we learned from the above coursework, literature on racial and gender bias and equity, and our experiences as largely tenured faculty who are regularly in a position to evaluate fellow scholars.

These recommendations are inspired by the work of *Bell et al., 2021*, *Berhe and Kim, 2019a*; *Berhe and Kim, 2019b*, *The University of Arizona Commission on the Status of Women, 2016*, *Okun, 2021* and *Itchuaqiyaq and Walton, 2021*. This project is for all academics who are in the position to evaluate Black scholars, and we recommend applying this rubric to letters written on behalf of any scholar in academia.

## Practical recommendations for writing anti-racist tenure and promotion letters

*Itchuaqiyaq and Walton, 2021* rightly point out that the "act of being called to review is also a call to power". This is doubly true for tenure review letters, and with that power comes a responsibility to hold both oneself and other power-holders (e.g., letter requesters, evaluation committees) accountable for anti-Black racism and to support the success of marginalized scholars. Here, we provide recommendations for non-Black academics writing letters for Black candidates; however, we believe that these recommendations apply more widely, both to other marginalized scholars and to letter writers of all identities. Based on *Okun, 2021* antidotes to 'white supremacy culture', we provide the following: (1) preparatory steps before writing a letter, (2) recommendations for writing a letter that recognizes Black excellence and contextualizes achievements within current academic culture, and (3) suggestions for what to do afterward to promote the voices of Black scholars and disrupt anti-Black racism in academia.

The list of recommendations incorporates feedback and reflects substantial contributions from other attendees and leaders of the 2021 Academics for Black Survival and Wellness training and from a diverse group of scholars in our professional networks. We emphasize that these recommendations for critical awareness and intentionality are important even when requested only to evaluate a candidate's scholarship, and are especially important the more marginalized intersecting identities the candidate holds.

### Before writing the letter

First, we encourage you to reflect on your various identities and background, including gender, race, class, sexual orientation, able-bodiedness, culture, ethnicity, religion and nationality. Reflect on both your privileged and marginalized statuses, and use this reflection both to set clear intentions for your letter and to gain a more holistic view on how your identities may impact your letter and your ability to evaluate different aspects of the candidate's dossier. If you are a white scholar writing a letter for a Black scholar, consider your own racial identity development (see *Helms, 2020*) and engage with scaffolded anti-racist resources as needed (*Stamborski et al., 2020*). In what ways does your identity align with the letter readers? In what way does your identity align with the subject of your letter? Why were you asked to write the letter? Clarify your positionality for yourself – what lens do you bring to this evaluation and how do your own identities and backgrounds shape your evaluative process? (See also: *Itchuaqiyaq and Walton, 2021*; *Clemons, 2019*; *UCLA Library, 2021a*;

*UCLA Library, 2021b*; Taylor Institute, 2022; *Derry, 2017*; *Darwin Holmes, 2020* Lacy, 2017).

Second, in order to gather relevant information and to encourage reflection on the part of the institution you are writing for, ask the person who requested the letter (e.g., the department chair) about the make-up and perspective of the department and institution (*Figure 1* top, box 2). How do they value and weigh research, teaching, and service? What are the faculty and student demographics? How do they account for, support, and evaluate diversity, equity, and inclusion (DEI) work? (See, for example, University of California, Berkeley's Rubric for Assessing Candidate Contributions to Diversity, Equity, Inclusion, and Belonging). How do they account for collaborative, ongoing, or community-facing endeavors? Ask for any additional information the person who requested the letter can supply about the candidate so that the candidate's often invisible and likely uncompensated DEI work can be included in your letter. If the institution or department does not explicitly value such scholarship and/or DEI work, consider how to frame such accomplishments within the institution's evaluation criteria.

As a final preparatory step (*Figure 1* top, box 3), research the candidate's CV, professional website, and public-facing social media to learn more about their influential work inside and outside of academia (e.g., DEI work, collaborative work, leadership). Consider the multiple ways that a research area has been impacted by the presence and contributions of the scholar and how to communicate the importance of work that is not traditionally valued by academia. It may be helpful to familiarize yourself with the embedded values of academia, through reading *Okun, 2021* or the work of Dr. Leslie Gonzales (e.g., *Gonzales and Waugaman, 2016*; *Gonzales and Núñez, 2014*), in order to recognize the limitations of traditional scholarship as the only currency of contribution to academia. This critical awareness will enable you to write a letter that speaks to those values while challenging tenure and promotion committees to expand their review beyond those values. It may be useful to follow discussions and groups on social media that are outside your immediate community, especially those that include scholars with identities other than your own. This can help to expand your understanding of what is valuable in academia and the hurdles Black scholars may face, and provide you with ideas on how to communicate this new knowledge to others.

### When writing the letter

Like we did in this paper, we suggest you include a positionality statement (see *Clemons, 2019*; *UCLA Library, 2021a*; *UCLA Library, 2021b*; *Taylor Institute, 2022*; *Derry, 2017*; *Darwin Holmes, 2020*; *Lacy, 2017*) or other description of your own backgrounds and experiences and how they shape how you are evaluating the candidate (*Figure 1* middle, box 1). It is typical to include a description of one's academic credentials, but we suggest you also be explicit in sharing the values you hold, as well as other identity factors that may influence your evaluation. Throughout, we recommend you use "I" statements that clarify the subjectivity of your assessments. While all assessment is subjective, an 'objective' meritocracy is a tantalizing illusion that is pervasive in academia.

In referring to the candidate, use language that amplifies their formal title or position (e.g., "Assistant Professor") rather than language that can detract from their credibility (e.g., "junior scholar," "early-career"). Throughout, we recommend you use "Dr." or another appropriate title rather than a first name (*Figure 1* middle, box 2), and that you consider how the scholar you are evaluating chooses to identify in public forums and refers to themselves professionally, and follow their lead.

The bulk of a tenure and promotion letter rests on the accomplishments of the candidate. Here, it is vitally important that you name *all* of the candidate's accomplishments (*Figure 1* middle, box 3). That is, in addition to mentioning traditional scholarship (papers, books, citations, invited talks, grants), you can expand your own – and the readers' – notion of what a scholarly accomplishment is. For example, you should call attention to: grant applications submitted (and re-submitted), symposia organized, spaces and classes created, leadership and service to the department and academic community, leadership to and education of the community outside of academia, creation of public policy and impact on public health, and participation in public relations or recruiting efforts. When possible, frame this as scholarship rather than service, because many of these achievements reflect the scholar's standing in the field.

We also encourage you to give context to how the scholar provides novel input, lays important groundwork, encourages you to think differently about your work, challenges the field with a different perspective, or moves the field forward. If their contributions are non-traditional and span

a wide range, you can highlight the range and quality of the various kinds of work they have completed (however, be careful to not emphasize effort over ability, e.g., "highly motivated"). Highlighting this broad range of accomplishments, even when they do not fit into the traditional metrics or definitions of scholarship, serves both to influence letter readers directly, and to present arguments to be used by internal advocates for the candidate.

For Black and other historically marginalized scholars in particular, recognize explicitly the candidate's experiences and characteristics that bring wisdom and perspective beyond chronological years and official titles (*Figure 1* middle, box 4). Those life experiences and identity characteristics (such as engaging in outreach programs, being an immigrant or international scholar, coming from a disadvantaged, under-resourced, or other path that has been historically underrepresented in academia) provide unique perspectives that enhance their environment. An anti-racist letter should also reflect a broad understanding of academic leadership as including collective and collaborative approaches. You can acknowledge the candidate's collaborations, the expertise they bring to group projects, and how they connect with the community within and beyond their institution.

An important aspect of contribution to scholarly work and discourse is mentoring, which is often disproportionately done by Black and other historically marginalized scholars. In your letter, elaborate on the candidate's mentorship (*Figure 1* middle, box 5): include the number of mentees (direct and indirect) that rely on the candidate for support and note the contributions and successes of the candidate's mentees and advisees. You can also include a statement about how the presence and/or work of the candidate creates a space of comfort for trainees and colleagues who do not traditionally feel welcome in academia (*Chaudhary and Berhe, 2020*). Acknowledge how the presence, actions, and intellectual contributions of the candidate draw developing scholars to the department/institution. Recognize and call attention to the time-consuming, but not consistently valued, public-relations work and other service work often required of marginalized scholars (*Gewin, 2020*).

We encourage you to add literature-supported encouragement for evaluators to account for systemic racism in academia (*Figure 1* middle, box 6). For example, you can include "Given the known racial disparities in grant funding (*Taffe*

*and Gilpin, 2021*) and publication rates (Lerback et al., 2020), and the epistemic exclusion of minoritized faculty (*Settles et al., 2022*),..." to provide context for your statements. It is important here to account for the many interpersonal and institutional barriers experienced by Black scholars, and to critique the devaluation of their work that provides tangible benefits to the university but is often unappreciated (*Rodríguez et al., 2015*). It may be helpful to explicitly state "even though the evaluation criteria do not consider [service/outreach/etc.], I include my assessment in this area given the vital importance of these contributions to the department and the field, and research on disproportionate service done by scholars of Color." We recommend emphasizing that achievement *in spite of* the systemic barriers enhances the value of the scholar's accomplishments rather than offering such barriers as a rationale for any potential perceived weaknesses.

We recommend you avoid (and ignore requests for) reductive comparisons to an arbitrary standard, a model/prototype, or a scholar at another institution (*Figure 1* middle, box 7). Such comparisons cannot account for the varied intersecting identities and experiences of different scholars.

Finally, check-in with yourself about your goals in taking an anti-racist approach to letter-writing, and ensure that you reflected them well in your letter (*Figure 1* middle, box 8). This approach is not about diluting the quality of tenured faculty or lowering the bar for promotion, but rather critiquing the devaluation of many types of scholarly contributions and recognizing the importance of such work both to the scholar's institution and to their field of study.

### After writing the letter

You can still do more. First, now that you are familiar with the work of this scholar in your field, make sure to cite them in your own work wherever appropriate (*Figure 1* bottom, box 1). Racial disparities in whose work is cited persist across a variety of disciplines (*Ray, 2018*; *Shirani, 2021*). As you might do for junior scholars you already know well, you can also reach out to the candidate to convey your appreciation of their scholarly work and offer specific support or mentorship, such as inviting them to give a talk at your department (*Figure 1* bottom, box 2). Do be attentive and respect their preferences if they decline your offer. If the junior scholar holds a marginalized identity and does take you up on the offer, educate yourself on how to mentor

them in a way that supports and respects their goals and values rather than suggest they adopt yours (a great starting point is *Fryberg and Gerken, 2012*; *Fryberg and Martínez, 2014*; *Martinez-Cola, 2020*).

Regardless of whether you connect directly, you can support the candidate through sponsorship and nominations (*Figure 1* bottom, box 3). For instance, you may identify an award for which you can nominate them based on all you learned while preparing your letter, or invite them to contribute a paper to a journal in which you are an editor. Additionally, you can broaden the network of people who are familiar with their scholarship, by inviting them to participate in a colloquium or to present their work at a conference you are organizing, including their work for discussion in a local journal club, and/or recommending their research group to graduate students and postdocs.

These actions on your part will not only benefit the scholar, they will also serve to educate your colleagues on the scholarship of this particular person, and on similar types of work they may be unfamiliar with (and might be called upon to evaluate in tenure and promotion letters in the future). Additionally, after following these suggestions for writing anti-racist tenure letters, consider expanding your anti-racism work by advocating for changes to the promotion criteria at your own institution drawing on the considerations outlined here.

Finally, as an intentionally anti-racist letter writer, we hope you will join a list of letter writers who have committed to using these guidelines (to join, fill in this form: https://forms.gle/4F4hBb9MNsrwdEDz6). This list (bit.ly/Tenure-Equity) will serve as a public resource for candidates and universities looking for anti-racist letter-writers.

We note that these 'after writing' steps are appropriate not only for scholars at the tenure/promotion career stage. Many early-career scholars who are marginalized in their field do not have a 'natural' mentoring and support network and could benefit from proactive acknowledgment and support.

## Conclusion

The guidance we provided above is based on our current best understanding of the nature of 'white supremacy culture' in academia and how to counteract it within the tenure recommendation letter. While our process is based on strong theoretical foundations (*Okun, 2021*) and shaped by the input of a diverse range of scholars and the leadership of the Academics for Black Survival and Wellness organization, these guidelines cannot perfectly apply to all situations and will need revision and reimagining over time. We recognize that the role of an external evaluator for a tenure candidate is one in which faculty have substantial freedom to adopt an anti-racist approach. We further suggest that scholars consider the many ways their anti-racist commitments can inform all aspects of their scholarly work (e.g., conducting peer review, see *Itchuaqiyaq and Walton, 2021*) and can be expanded to bring about systemic changes in their institutions.

To be sure, anti-racist tenure letters may be met with resistance and even backlash by tenure committees, as many academics are (implicitly) committed to maintaining power structures that are familiar to them, and that confer them with outsized power and privilege. We suggest to directly rebut, in your letter, what might traditionally be considered 'weaknesses' in the applicant's file, by explicitly addressing why you do not consider these as weaknesses. This can be done throughout your letter (and we have suggested specific ideas for how to do this in the above recommendations), as well as in an explicit rebuttal paragraph, as we are doing here. Such resistance-anticipating arguments will provide much needed ammunition for other advocates involved in the tenure process at the candidate's home institution.

Unfortunately, our suggestions cannot ameliorate the persistent opportunity gaps and discrimination that Black scholars face in their educational and professional careers. We offer these recommendations as one avenue for resisting 'white supremacy culture' in academia with the understanding that such resistance must be accompanied by systemic reforms. Even as some institutions introduce diversity, equity and inclusion (DEI) requirements for tenure or forge new DEI-focused pathways to tenure, the success of such projects in creating an inclusive and equitable academic culture will require openly acknowledging racism in academia (*Gosztyla et al., 2021*), taking a proactive approach to an inclusive workplace culture and retaining Black scholars (*National Academies of Sciences, Engineering, and Medicine, 2022*), and introducing changes to tenure evaluation policies (e.g. recognizing the myriad biases in student evaluations of teaching) and practices (e.g, nomination and selection of letter writers, instructions provided to letter writers, how evaluators are trained).

While individual letter writers have substantial input in the tenure process, tenure decisions ultimately rest with deans, provosts, presidents, and boards. These power-holders are uniquely situated to reimagine and advocate for a tenure process that centers equity, inclusion, and justice rather than a process that reproduces 'white supremacy culture'. In addition to writing anti-racist letters, we encourage you to advocate that the power-holders in your own institution reimagine tomorrow's academia and proceed with haste to make that dream a reality. It is in their – and all of our – hands.

## Acknowledgements

We gratefully acknowledge the creators and participants of the 2020, 2021 and 2022 Academics for Black Survival and Wellness trainings, particularly the teachings and inspirational anti-racist activism of Dr. Della Mosley, Dr. Carlton Green, and Adania Flemming. We look forward to furthering our learning at future A4BL trainings and encourage new participants to join us. This manuscript is the culmination of work by a working group that formed in the 2021 training. We thank Aditi Jayarajan and Dr. Caroline Storer for their recommendations and support in the initial stages of this project. We are also grateful to our reviewers and editors at eLife for their excellent suggestions for improving this manuscript. We acknowledge that the lands we inhabit were previously occupied by indigenous peoples who were, in many cases, forcibly removed, and that the physical academic spaces in which we work and from which we generated this collaborative product were built, in large part, using uncompensated and often forced labor of Black people.

### Group author details

**The A4BL Anti-racist Tenure Letter Working Group**
**Pearis Bellamy**: Department of Psychology, University of Florida, Gainesville, United States; **Christy L Haynes**: Department of Chemistry, University of Minnesota, Minneapolis, United States; **Laura E Martin**: Department of Population Health and Hoglund Biomedical Imaging Center, University of Kansas Medical Center, Kansas City, United States; **Scott P Mirabile**: Department of Psychology, St. Mary's College of Maryland, Saint Marys City, United States; **Yael Niv**: Department of Psychology and Princeton Neuroscience Institute, Princeton University, Princeton, United States; **Jacqueline K Rose**: Department of Psychology, Western Washington University, Bellingham, United States; **Rachel A Ross**: Departments of Neuroscience, Psychiatry, and Medicine, Albert Einstein College of Medicine, New York City, United States

pearisbellamy@ufl.edu;chaynes@umn.edu;lmartin2@kumc.edu;spmirabile@smcm.edu;yael@princeton.edu; rosej8@wwu.edu;rachel.ross@einsteinmed.edu

*Author contributions: The A4BL Anti-racist Tenure Letter Working Group, Writing – original draft, Writing – review and editing*

*Competing interests:* The authors declare that no competing interests exist.

## Funding

No external funding was received for this work.

## Decision letter and Author response

Decision letter https://doi.org/10.7554/eLife.79892.sa1
Author response https://doi.org/10.7554/eLife.79892.sa2

## Data availability

No data were generated for this study.

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
