## [Decision Letter]

**Decision letter after peer review:**

Thank you for submitting your article "A guide for writing anti-racist tenure and promotion letters" to *eLife* for consideration as a Feature Article. Your article has been reviewed by two peer reviewers, and the evaluation has been overseen by two members of the *eLife* Features Team (Julia Deathridge and Peter Rodgers). The following individual involved in the review of your submission has agreed to reveal their identity: Isis Settles.

We have drafted this decision letter to help you prepare a revised submission.

Summary

This article provides a well-thought-out set of recommendations for how senior academics can write anti-racist tenure letters that better recognize the contributions of Black scholars. The authors – a group of researchers who took part in an Academics for Black Survival and Wellness workshop – advise what academics should do before writing the letter, what should (and should not be) included, and what letter writers can do after to continue to support faculty from marginalized groups. The manuscript brings much-needed attention to the power senior academics hold when writing these documents and avoids deficit framing, which is critical for an anti-racist approach. If widely adopted, these guidelines could have an incredible impact and lead to a more inclusive academic culture. However, there are a few points that need to be addressed before the article can be accepted for publication.

Essential Revisions

1) The authors introduce white supremacy culture as a cultural force that requires disrupting and introduce reconsidering tenure letters as a mechanism to do so. I'm concerned that this section seems too abstract, particularly given the concrete suggestions provided later. Could the authors ground this section further? For example, the last sentence currently reads: "We have a right to shape our shared culture and to promote an academic community that recognizes and benefits from the wisdom of all people who participate in academia." Can I suggest changing this to: "Those of us committed to disrupting this implicit and harmful culture have a right and obligation to actively promote an academic community that recognizes and benefits from the expertise of all people who participate in academia. One way to accomplish this on an individual level is to reconsider how we write letters of assessment, including tenure letters, so that they embody the cultural shift we'd like to see."

2) In the Okun (2021) framework, perfectionism includes the idea that there is one way of doing things and objectivity is possible. I wonder if there is value in tying this to the positivist and post-positivist values of science that have been adopted by many fields. I think the values of science are invisible to most scholars so it could be helpful to discuss how these are just types of epistemologies of science but there are other knowledge systems that exist and are used.

3) On p. 4, you note that writers should "Consider how to communicate the importance of such work that is not traditionally valued by academia." I wonder if you have ideas about how letter writers can educate themselves in this area – both in the traditional values of academia (Leslie Gonzales at Michigan State University has done work in this area) and the specific contributions of a scholar's work. For individuals doing more traditional or mainstream work, they may not have considered the contributions of work on the margins which will make it challenging for them to communicate those things effectively.

4) On page 5 (lines starting at 183) where you talk about accomplishments, I suggest two additions. First, broadening notions of impact and contribution beyond the usual metrics to include things like impact on public policy, public health, community engagement, teaching, and education, etc. Second, when scholars do public or engaged scholarship, it is often considered to be service. Where possible, writers should frame this as scholarship because it is and that's where it usually counts most for promotion.

5) Although you recommend that writers not feel tied to only addressing the criteria shared by the institution requesting the letter (page 5 – line 222), I would state this more explicitly. In my letters for scholars from marginalized groups, I often will say something like "even though the evaluation criteria don't consider X (e.g., service), I include my assessment in this area given the research on disproportionate service labor done by scholars of color."

6) Although I appreciate the framing of letter writers as individuals who may not be able to change the broader system, I encourage the authors to include some advice about making structural changes in the "after the letter" section. For example, individuals could seek to educate their colleagues about the scholarship of the person they evaluated and similar types of work (through invited speakers, journal club selections, etc.). You have suggested that they bring in scholars as speakers to benefit the scholar's career, but I'm suggesting using these activities to educate others who may find themselves in a position to also write letters for scholars from marginalized groups. Individuals could also work to change their internal promotion criteria through their roles as senior scholars. This combination of individual and structural changes may allow for change to (hopefully) happen on a wider and faster scale.

7) I think the authors have to confront the very real possibility of resistance and backlash these letters may come up against. In the sections called Before Writing the Letter and When Writing the Letter, I'd recommend adding suggestions for how to directly rebut what might be considered "weaknesses" in the applicant's file by detractors insistent upon maintaining academia's current culture. Explicitly addressing why the letter writer does not consider these weaknesses provides much-needed ammunition for other advocates involved in the tenure process at the applicant's home institution.

8) Please consider including the following citations:

a) The National Academies of Sciences, Engineering, and Medicine: Promotion, Tenure, and Advancement through the Lens of 2020: The next normal for the advancement of tenure and non-tenure-track faculty (https://nap.nationalacademies.org/catalog/26405/promotion-tenure-and-advancement-through-the-lens-of-2020-proceedings) – this series of papers may offer additional ideas for advice for equitable evaluations

b) Intersectional inequalities in science (https://doi.org/10.1073/pnas.2113067119) talks about gender and racial bias in citation practices

c) Gonzales, L. D. (2018). Subverting and minding boundaries of knowledge production in academe: The intellectual work of women. The Journal of Higher Education, 89(3) 1-25. – Describes values in academia from a historical perspective.

---

## [Author Response]

Essential Revisions1) The authors introduce white supremacy culture as a cultural force that requires disrupting and introduce reconsidering tenure letters as a mechanism to do so. I'm concerned that this section seems too abstract, particularly given the concrete suggestions provided later. Could the authors ground this section further? For example, the last sentence currently reads: "We have a right to shape our shared culture and to promote an academic community that recognizes and benefits from the wisdom of all people who participate in academia." Can I suggest changing this to: "Those of us committed to disrupting this implicit and harmful culture have a right and obligation to actively promote an academic community that recognizes and benefits from the expertise of all people who participate in academia. One way to accomplish this on an individual level is to reconsider how we write letters of assessment, including tenure letters, so that they embody the cultural shift we'd like to see."

Thank you for identifying that this section was too abstract. We have adopted your suggestion verbatim, but also rewrote the section introducing White Supremacy Culture and tried to make it more explicit, give a concrete example, and be more prescriptive about what is needed:

“These “White Supremacy Culture” values (Okun, 2021) are not a necessity in academia. Most of us, having been trained and steeped in this culture for years, may not even recognize these invisible but ever-present “rules of the game” despite the fact that these rules limit creativity and inclusion. Naming these values as ones we have adopted makes clear that they are not axioms of academic culture. There are alternatives. Those of us committed to disrupting theis implicit and harmful culture have a right and obligation to actively promote an academic community that recognizes and benefits from the expertise of all people who participate in academia. One way to accomplish this on an individual level is to reconsider how we write letters of assessment, including tenure letters, so that they embody the cultural shift we'd like to see.

In particular, the academic profession was developed to have strict boundaries about who is allowed to be a part of the profession and what work is valued. Western scientific approaches (i.e.g., objectivity, linearity, generalizability, and detachment) have long been the standard for evaluating academic rigor and membership into this highly selective profession. Thus, Black scholars engaging in community- focused research and racial- justice work are often evaluated negatively because their work is viewed as not being objective and their advocacy is viewed as problematic (Gonzales, 2018). Indeed, Black scholars are overrepresented in topics related to racial discrimination and African studies but often cited less than other racial groups of scholars (Kozlowski et al., 2022). Therefore, in evaluation processes, like tenure and promotion, their research is often already being undervalued by evaluators enmeshed in “White Supremacy Culture.” Anti-racist tenure and promotion letters provide an avenue of intervention and advocacy to challenge the exclusionary and harmful aspects of academia. In evaluating scholarship that does not necessarily conform to “White Supremacy Culture” values, we must recognize that our personal biases influence both our scientific practice and our tendency to uphold these values of scientific pursuit. If we are to move beyond these exclusionary practices, we must recognize these biases in all of our academic practices and value other knowledge systems beyond that of the ‘traditional’ epistemology of science.” (pp. 2)

2) In the Okun (2021) framework, perfectionism includes the idea that there is one way of doing things and objectivity is possible. I wonder if there is value in tying this to the positivist and post-positivist values of science that have been adopted by many fields. I think the values of science are invisible to most scholars so it could be helpful to discuss how these are just types of epistemologies of science but there are other knowledge systems that exist and are used.

We wholeheartedly agree, and our impetus in introducing Okun’s framework is to clarify that there are alternatives – we can adopt other values, we can value other ways of knowing, and other knowledge systems. We have now tried to say this more explicitly when introducing White Supremacy Culture (see the text reproduced above). In particular, our call to “recognize that our personal biases influence both our scientific practice and our tendency to uphold these values of scientific pursuit. If we are to move beyond these exclusionary practices, we must recognize these biases in all of our academic practices and value other knowledge systems beyond that of the ‘traditional’ epistemology of science.” is trying to explicitly name the work that must be done, in terms of being aware of these biases, recognizing their possible effects, and working to counteract them, as per the post-positivism framework (we decided to not explicitly name post-positivism as readers may not be familiar with that term).

3) On p. 4, you note that writers should "Consider how to communicate the importance of such work that is not traditionally valued by academia." I wonder if you have ideas about how letter writers can educate themselves in this area – both in the traditional values of academia (Leslie Gonzales at Michigan State University has done work in this area) and the specific contributions of a scholar's work. For individuals doing more traditional or mainstream work, they may not have considered the contributions of work on the margins which will make it challenging for them to communicate those things effectively.

You are correct, and this is a tough one. Most of us have been learning about these issues for years – from social media, as well as more formal training (A4BL) and readings. In the spirit of acknowledging multiple ways of learning and knowing, we have added pointers both to scholarly work (thank you for pointing out Leslie Gonzalez, who we now refer to) as well as other sources of knowledge (Twitter, Facebook):

“As a final preparatory step (Figure 1 top, box 3), research the candidate’s CV, professional website, and public-facing social media to learn more about the candidate’s influential work inside and outside of academia (e.g., DEI work, collaborative work, leadership). Consider the multiple ways that a research area has been impacted by the presence and contributions of the scholar and how to communicate the importance of any work that is not traditionally valued by academia. It may be helpful to familiarize yourself with the embedded values of academia, through reading Okun (2021) or the work of Dr. Leslie Gonzales (e.g., Gonzales and Waugman, 2016; Gonzales and Núñez, 2014), in order to recognize the limitations of traditional scholarship as the only currency of contribution to academia. This critical awareness will enable you to write a letter that speaks to those values while challenging tenure and promotion committees to expand their review beyond those values. It may be useful to follow discussions and groups on social media (Twitter, Facebook) that are outside your immediate community, especially of scholars with identities other than your own. This can help to expand your understanding of what is valuable in academia and the hurdles Black scholars may face, and provide you with ideas on how to communicate this new knowledge to others.” (pp. 5)

Inspired by your suggestion, and in response to comment 7 below, we also note, in a later section, how this self-education can translate to wording in the letter that will help advocates in the committee argue for inclusion of not-traditionally-valued work and achievements as part of the candidate’s recognized scholarship:

“Highlighting this broad range of accomplishments, even when they do not fit into the traditional metrics or definitions of scholarship, serves both to influence letter readers directly, and to present arguments to be used by internal advocates for the candidate.” (pp. 5)

4) On page 5 (lines starting at 183) where you talk about accomplishments, I suggest two additions. First, broadening notions of impact and contribution beyond the usual metrics to include things like impact on public policy, public health, community engagement, teaching, and education, etc. Second, when scholars do public or engaged scholarship, it is often considered to be service. Where possible, writers should frame this as scholarship because it is and that's where it usually counts most for promotion.

Thank you for these excellent suggestions. We have now added these to the text as follows:

“The bulk of a tenure and promotion letter rests on the accomplishments of the candidate. Here, it is vitally important that you name *all* of the candidate’s accomplishments (Figure 1 middle, box 3). That is, in addition to mentioning traditional scholarship (papers, books, citations, invited talks, grants), you can expand your own – and the readers’ – notion of what a scholarly accomplishment is. Grant applications submitted (and re-submitted), symposia organized, spaces and classes created, leadership and service to the department, profession, and academic community, leadership to and education of the community outside of academia, creation of public policy and impact on public health, and participation in public relations or recruiting efforts are all important accomplishments that you should call attention to. When possible, frame this as scholarship, rather than service, because many of these achievements reflect the scholar’s standing in the field.” (pp. 5)

5) Although you recommend that writers not feel tied to only addressing the criteria shared by the institution requesting the letter (page 5 – line 222), I would state this more explicitly. In my letters for scholars from marginalized groups, I often will say something like "even though the evaluation criteria don't consider X (e.g., service), I include my assessment in this area given the research on disproportionate service labor done by scholars of color."

This is a great way of framing your comments in letters. We have now added this sentence as part of step 6 “during”, on page 7:

“It is important here to account for the many interpersonal and institutional barriers experienced by Black scholars, and to critique the devaluation of their work that provides tangible benefits to the university but is often unappreciated (e.g., Rodríguez, Campbell, and Pololi, 2015). It may be helpful to explicitly state “even though the evaluation criteria do not consider [service/outreach/etc.], I include my assessment in this area given the vital importance of these contributions to the department and the field, and research on disproportionate service done by scholars of Color.” (pp. 7)

6) Although I appreciate the framing of letter writers as individuals who may not be able to change the broader system, I encourage the authors to include some advice about making structural changes in the "after the letter" section. For example, individuals could seek to educate their colleagues about the scholarship of the person they evaluated and similar types of work (through invited speakers, journal club selections, etc.). You have suggested that they bring in scholars as speakers to benefit the scholar's career, but I'm suggesting using these activities to educate others who may find themselves in a position to also write letters for scholars from marginalized groups. Individuals could also work to change their internal promotion criteria through their roles as senior scholars. This combination of individual and structural changes may allow for change to (hopefully) happen on a wider and faster scale.

Yes! Ideally all this would happen. We would love it if these recommendations had impact beyond individual letters and careers, to more systemic change and wider education. This is a great suggestion. We have now added a paragraph that explicitly discusses this at the end of the “after the letter” section, in addition to the relevant text we already had in the conclusion section:

“These actions on your part will not only benefit the scholar, they will also serve to educate your colleagues on the scholarship of this particular person, and on similar types of work they may be unfamiliar with (and might be called upon to evaluate in tenure and promotion letters in the future). Indeed, after following these suggestions for writing anti-racist tenure letters, consider expanding your anti-racism work by advocating for changes to the promotion criteria at your own institution drawing on the considerations outlined here.“ (pp. 7)

“We recognize that the role of an external evaluator for a tenure candidate is one in which faculty have substantial freedom to adopt an anti-racist approach. We further suggest that scholars consider the many ways their anti-racist commitments can inform all aspects of their scholarly work (e.g., conducting peer reviews, see Itchuaqiyaq and Walton, 2021), and can be expanded to bring about systemic changes in their institutions.” (pp. 8)

Previous relevant text in our conclusions section:

“Even as some institutions introduce diversity, equity and inclusion (DEI) requirements for tenure or forge new DEI-focused pathways to tenure, the success of such projects in creating an inclusive and equitable academic culture will require openly acknowledging racism in academia (Gosztyla et al., 2021), taking a proactive approach to an inclusive workplace culture and to retaining Black scholars (National Academies of Sciences, Engineering, and Medicine, 2022), and instituting changes to tenure evaluation policies (e.g., recognizing the importance of community-based scholarship, recognizing the myriad biases in student evaluations of teaching) and practices (e.g., nomination and selection of letter writers, instructions provided to letter writers, how evaluators are trained). While individual letter writers have substantial input in the tenure process, tenure decisions ultimately rest with deans, provosts, presidents, and boards. These power-holders are uniquely situated to reimagine and advocate for a tenure process that centers equity, inclusion, and justice rather than a process that reproduces “White Supremacy Culture.” In addition to writing anti-racist letters, we encourage you to advocate that the power-holders in your own institution reimagine tomorrow’s academia and proceed with haste to make that dream a reality. It is in their – and all of our – hands.” (pp. 8)

7) I think the authors have to confront the very real possibility of resistance and backlash these letters may come up against. In the sections called Before Writing the Letter and When Writing the Letter, I'd recommend adding suggestions for how to directly rebut what might be considered "weaknesses" in the applicant's file by detractors insistent upon maintaining academia's current culture. Explicitly addressing why the letter writer does not consider these weaknesses provides much-needed ammunition for other advocates involved in the tenure process at the applicant's home institution.

This is, in many senses, the elephant in the room. We have intentionally chosen to write from a positive perspective that assumes that letter writers (or at least, those reading our paper) are well-intentioned but ill-equipped to act on their intentions. However, not everyone is, and as pointed out implicitly in some of the above comments, even if the letter-writer is doing anti-racist work, the committee reading the letter may not be on board with these values (yet). As such, the letter writer themselves may have to confront the very real possibility of resistance and backlash.

To address this, and through that model to letter writers what they can do as well, we have added what is known as a “To be sure” paragraph in op eds – a direct address to the most commonly heard criticism of what we are arguing for. If this is not what you had in mind, or if you have additional suggestions for how to address resistance and backlash, we are happy to add to this:

“To be sure, anti-racist tenure letters may be met with resistance and even backlash by tenure committees, as many academics are (implicitly) committed to maintaining power structures that are familiar to them, and that confer them with outsized power and privilege. We suggest to directly rebut, in your letter, what might traditionally be considered "weaknesses" in the applicant's file, by explicitly addressing why you do not consider these as weaknesses. This can be done throughout your letter (and we have suggested specific ideas for how to do this in the above recommendations), as well as in an explicit rebuttal paragraph, as we are doing here. Such resistance-anticipating arguments will provide much-needed ammunition for other advocates involved in the tenure process at the candidate's home institution.” (pp. 8)

In addition, in reply to the above comments, in several places we have buffered up suggestions for explicitly addressing why issues that are traditionally considered weaknesses, are not, and how this can help internal advocates.

8) Please consider including the following citations:a) The National Academies of Sciences, Engineering, and Medicine: Promotion, Tenure, and Advancement through the Lens of 2020: The next normal for the advancement of tenure and non-tenure-track faculty (https://nap.nationalacademies.org/catalog/26405/promotion-tenure-and-advancement-through-the-lens-of-2020-proceedings) – this series of papers may offer additional ideas for advice for equitable evaluationsb) Intersectional inequalities in science (https://doi.org/10.1073/pnas.2113067119) talks about gender and racial bias in citation practicesc) Gonzales, L. D. (2018). Subverting and minding boundaries of knowledge production in academe: The intellectual work of women. The Journal of Higher Education, 89(3) 1-25. – Describes values in academia from a historical perspective.

Thank you for these excellent suggestions. We have now incorporated citations (b) and (c) in our introduction and discussion of White Supremacy Culture, as a practical example of how this culture harms Black scholars and how anti-racist tenure and promotion practices may help, and cite (a) in the conclusion paragraph:

In particular, the academic profession was developed to have strict boundaries about who is allowed to be a part of the profession and what work is valued. Western scientific approaches (i.e.g., objectivity, linearity, generalizability, and detachment) have long been the standard for evaluating academic rigor and membership into this highly selective profession. Thus, scholars of Color engaging in community- focused research and racial- justice work are often evaluated negatively because their work is viewed as not being objective and their advocacy is viewed as problematic (Gonzales, 2018). Indeed, Black scholars are overrepresented in topics related to racial discrimination and African studies but often cited less than other racial groups of scholars (Kozlowski et al., 2022). Therefore, in evaluation processes, like tenure and promotion, their research is often already being undervalued by evaluators enmeshed in “White Supremacy Culture.” Anti-racist tenure and promotion letters provide an avenue of intervention and advocacy to challenge the exclusionary and harmful aspects of academia. In evaluating scholarship that does not necessarily conform to “White Supremacy Culture” values, we must recognize that our personal biases influence both our scientific practice and our tendency to uphold these values of scientific pursuit. If we are to move beyond these exclusionary practices, we must recognize these biases in all of our academic practices and value other knowledge systems beyond that of the ‘traditional’ epistemology of science.” (pp. 2)

“Even as some institutions introduce diversity, equity and inclusion (DEI) requirements for tenure or forge new DEI-focused pathways to tenure, the success of such projects in creating an inclusive and equitable academic culture will require openly acknowledging racism in academia (Gosztyla et al., 2021), taking a proactive approach to an inclusive workplace culture and to retaining Black scholars (National Academies of Sciences, Engineering, and Medicine, 2022), and instituting changes to tenure evaluation policies (e.g., recognizing the importance of community-based scholarship, recognizing the myriad biases in student evaluations of teaching) and practices (e.g., nomination and selection of letter writers, instructions provided to letter writers, how evaluators are trained).” (pp. 8)